# Risk Determinants of Sexual Behaviors: Dating Apps, History of Sexually Transmitted Infections, Substance Use, and Pornography Consumption in Health Science Students

**DOI:** 10.3390/nursrep15030083

**Published:** 2025-02-28

**Authors:** María Naranjo-Márquez, Anna Bocchino, Ester Gilart, Eva Manuela Cotobal-Calvo, Fortuna Procentese, José Luis Palazón-Fernández

**Affiliations:** 1Salus Infirmorum Nursing Center, University of Cádiz, 11001 Cádiz, Spain; marianaranjo@salusinfirmorumcadiz.com (M.N.-M.); evamanuela.cotobal@ca.uca.es (E.M.C.-C.); jluis.palazon@uca.es (J.L.P.-F.); 2Department of Nursing and Physiotherapy, University of Cádiz, 11001 Cádiz, Spain; ester.gilart@uca.es; 3Department of Humanities, University Federico II of Naples, 80126 Naples, Italy; fortuna.procentese@unina.it

**Keywords:** sexually transmitted infections, sexual risk behavior, health sciences university students, dating apps, drug and pornographic material consumption

## Abstract

**Background**: Since 2020, there has been a significant increase in sexually transmitted infections (STIs), especially in young people, and these include syphilis, gonorrhea, chlamydia, and lymphogranuloma venereum, which are often asymptomatic but with the potential for transmission. In addition, certain risk behaviors, such as the use of dating apps, pornography, and substance use, reduce adherence to barrier methods, especially in men, thus facilitating the spread of these infections. **Methods**: This observational, cross-sectional study aimed to explore the relationship between the use of dating apps, drug and pornographic material consumption, and STI history in university students of the health sciences. **Results**: The sample consisted of 730 participants. The results indicated that individuals who identified as gay, lesbian, or bisexual reported significantly higher rates of dating app use, drug use, and pornography consumption compared to those who identified as heterosexual. Also, these groups showed lower adherence to contraceptive and protective methods. A statistical analysis revealed a relationship between the use of dating apps and increased sexual risk behaviors, suggesting that the accessibility of these platforms could influence the frequency and type of sexual contact. **Conclusions**: The increase in the prevalence of STIs in recent years has highlighted the urgency of strengthening prevention and sexual health promotion strategies, especially in young and high-risk populations. This study emphasizes the need for early and targeted interventions in high-risk groups to reduce the incidence of STIs and promote responsible sexual health practices.

## 1. Introduction

Sexually transmitted infections (STIs) are a group of diseases that are primarily spread through sexual contact and represent a global public health problem. According to data compiled by the WHO, in 2020, more than 374 million new cases of STIs were reported worldwide [1].

STIs can be transmitted by pathogens through blood, semen, vaginal fluids, and body fluids [2]. Some STIs may be asymptomatic [1], meaning that they do not manifest obvious clinical symptoms, although they can be transmitted to others.

In 2022, the Spanish Association of Pediatrics warned of an upward trend in the incidence of STIs in Spanish adolescents, with the prevalence having doubled in number in the 15–19 age group between 2016 and 2019 [3]. Furthermore, in the latest report on STIs published in 2024, the National Epidemiological Surveillance Network reported that among adults, the age group between 20 and 24 years old presented with the highest incidence of gonococcal infection and Chlamydia trachomatis, in contrast to syphilis and lymphogranuloma venereum, which were the most common in the 25–34 age group. These four diseases are notifiable STIs [4]. In the Autonomous Community of Andalusia, data recorded by the Andalusian Health System on STIs show an 81.98% increase in cases in 2023 compared to 2022 [5].

More than 30 pathogens that are transmitted through sexual contact have been identified, 8 of which are linked to the most common sexually transmitted infections such as syphilis, gonorrhea, chlamydia, trichomoniasis, hepatitis C (HCV), herpes simplex virus (HSV), human immunodeficiency virus (HIV), and human papillomavirus (HPV) [1]. It is important to note that the first four infections are curable; the other four are viral infections, and their transmission has long-term implications in terms of reproductive health, quality of life, and mortality. Cervical cancer has been the fourth leading cause of cancer deaths in women, with about 660,000 new cases and about 350,000 deaths worldwide [6], and it represents about 80% of HPV’s attributable cancer burden [7].

Although vaccination strategies for HPV are now in place in Spain [8], other STIs (such as syphilis or gonococcal infection) are seeing major spikes [4].

In addition, having multiple STIs can complicate the treatment and management of these diseases, increasing the likelihood of contracting HIV. Importantly, STIs can also be transmitted during pregnancy [1,9], posing a significant risk to the health of the baby.

In the United Nations 2030 agenda, the eradication of diseases such as AIDS and other STIs has been identified as one of the sustainable development goals [10,11].

Given the potential severity and public health implications associated with STIs, it is crucial that sexually active people are screened regularly. This recommendation is particularly emphasized in individuals that engage in sexual risk behaviors (SRB), as some situations increase the likelihood of transmission.

Several factors have been related to STI transmission, including drug [12] and pornography [13,14] consumption, the use of dating apps [15,16], the number of partners [17,18], religion [19,20], and non-use or inappropriate use of contraception [21,22].

Multiple studies have clarified how these factors interact with STIs, such as the following: In Mexico, a study conducted on young people reported that the probability of acquiring an STI while using drugs was 2.8 times higher. The most used drug was marijuana, followed by cocaine, which is chosen for its effects. Its use during sexual intercourse was associated with increased sensations, prolonged orgasm, and increased rates of SRB [12]. Searching for pornographic material on the internet has been associated with reduced condom use, which is directly related to STIs, especially in men [13]. Pornography use was also associated with having multiple sexual partners [14]. It cannot be confirmed that there is a direct relationship between the use of dating apps and STIs, but it can be confirmed that these users have a greater number of partners and use condoms less frequently, due to the immediacy of finding people to have sex with [16,23]. Studies carried out in both female and male populations have shown a direct relationship between STIs and multiple partners [17,18]. Religion often limits sexual practices with regard to vaginal penetration but could favor anal and oral practices, as well as the viewing of pornography [19,20]. In Spain, in 2020, only 31.3% of women of childbearing age used a condom [21]. A study carried out in Galicia found significant correlations between the use of emergency contraception and the prevalence of syphilis, hypothesizing an increase in STIs due to the free availability of and easy access to emergency contraception in primary health care centers [22]. However, the existing literature is heterogeneous, and methodological problems make it difficult to draw reliable conclusions. This highlights a significant lack of studies specifically examining the relationship between sexual orientation, drug use, and dating app use. The lack of data on the relationship between sexual orientation, drug use, and dating app use emphasizes the need for standardized measures and representative samples in future studies [24,25,26].

Therefore, the main objective of this study is to describe the relationship between socio-demographic characteristics, the history of STIs, the use of dating apps, the consumption of pornographic material, and drug use in a sample of health sciences university students.

## 2. Materials and Methods

### 2.1. Participants

Using the G-Power 3.1 program, a minimum sample size of *n* = 252 was calculated for an effect size of 0.25, alpha = 0.05 and a power of 0.95. Nonetheless, an attempt was made to recruit as many students as possible; therefore, the final sample included 730 students. The median age of the participants was 21 years, and all of them were studying a degree related to health sciences.

### 2.2. Instruments

We used an ad hoc demographic questionnaire. It contained questions related to age, gender, the health science program studied and academic year level, sexual orientation, religion, use of contraceptive methods, history of sexually transmitted diseases, marital status, use of dating apps and pornographic material, and drug and alcohol consumption.

### 2.3. Procedure

This descriptive cross-sectional observational study was conducted in 2023 using convenience sampling. Participants were recruited from the available cohort of health sciences students, and data collection was performed by convenience sampling using the online questionnaire tool Google Forms. For this purpose, we worked with collaborators in different health sciences faculties. The online questionnaire was disseminated through them, making it available to the greatest number of health sciences students possible. Students who agreed to participate and signed the informed consent form completed the questionnaire, whose estimated response time was approximately 10 min. Participation was completely voluntary, with students allowed to withdraw at any time if they wished. The confidentiality and anonymity of the responses was guaranteed, ensuring that only the research team had access to the data collected. The inclusion criteria were as follows: being a student of any health sciences program at the time of the survey, being an adult (over 18), and providing informed consent.

This study was conducted in accordance with the 2013 Declaration of Helsinki (seventh revision, 64th meeting, Fortaleza) and the Organic Law 3/2018 of December 5, on the protection of personal data and guarantees of digital rights in Spain. Prior to data collection, the participants were informed about the objectives of the study, that their participation was completely voluntary (they could leave the study at any time if they did not feel comfortable for any reason), and that completion of the questionnaire implied they gave informed consent to participate. It was emphasized that the responses were anonymous and confidential in order to promote honesty. To ensure the privacy of the participants, only the research team had access to the data collected. The academic institutions granted approval for this study.

### 2.4. Data Analysis

The quantitative variable (age) was expressed as median and interquartile range. Categorical variables were described as frequencies and percentages.

Relationships between categorical variables were studied using the Chi-square test.

The data were analyzed using the IBM SPSS version 26.0 software for Windows (IBM Corp., Armonk, NY, USA). For all tests, *p*-values ≤ 0.05 were considered significant.

## 3. Results

The sociodemographic characteristics, use of dating apps, and drug and pornography consumption of the 730 people in the sample, as well as the results of the χ^2^ test to assess the relationships between these variables, are presented in Table 1, Table 2 and Table 3.

### 3.1. Socio-Demographic Variables

Seventy percent of the participants were female. The median age was 21 years, with a range of 17 to 52 years (RIC 20 to 22 years). Almost half of the participants were studying nursing (46.7%) and were in their second year of studies (48.1%). The majority (56.9%) had a partner, and the vast majority of those who had a partner were in an exclusive or traditional relationship (95.8%), although only 9.7% were living as a couple. In terms of religious affiliation, 69.2% identified themselves as Catholic, 27.8% as agnostic, and the rest as Muslim or Protestant.

#### 3.1.1. Sexual Orientation

A total of 84.4% of the participants identified themselves as heterosexual, 9.3% identified themselves as bisexual, 4.4% identified themselves as gay or lesbian, and 1.9% declared themselves asexual or preferred not to state their sexual orientation (Table 1). A significant relationship (*p* < 0.05) was observed between sexual orientation and gender, religion, type of relationship, use of contraceptives, use of dating apps, consumption of pornographic material, and drug use before or during sexual intercourse (Table 2).

Among men, 82.8% reported themselves as heterosexual, 9.6% reported themselves as gay or lesbian, and 6.2% reported themselves as bisexual. Among women, these percentages are 85.9% for heterosexuals, and 2.0% and 6.2% for those who identified as gay or lesbian and bisexual, respectively.Although the majority of the participants identified themselves as Catholic, the percentages varied according to sexual orientation (72.7% among heterosexuals, 65.6% among individuals who identified as gay or lesbian, and 45.6% among bisexuals).In total, 97.2% of heterosexuals were in an exclusive relationship, compared to 95% of individuals who identified as gay or lesbian and 87.5% of bisexuals.In terms of contraceptive use, 82.5% of heterosexuals, 71.2% of bisexuals, and 53.1% of individuals who identified as gay or lesbian used contraceptives.The use of dating apps was reported by 59.4% of individuals who identified as gay or lesbian, 44.1% of bisexuals, and 15.6% of heterosexuals.Consumption of pornographic material was 53.1% among individuals who identified as gay or lesbian (52.6% 1–3 a week), 47.1% among bisexuals (66.7% 1–3 times per week), and 32.3% among heterosexuals (78.7% 1–3 times per week).Drug use was 71.9% among individuals who identified as gay or lesbian (90.9% 1–3 times a week and 9.4% before or during sexual intercourse), 60.3% among bisexuals (74.4% 1–3 times a week and 7.4% before or during sexual intercourse), and 58.3% among heterosexuals (90.8% 1–3 times a week and 1.6% before or during sexual intercourse).

#### 3.1.2. Contraceptive Use

In total, 80.1% of the participants reported that they used contraception (Table 1). Among these, condoms (63.5%) and contraceptive pills (26.3%) were the most commonly used. Contraceptive use was significantly related (*p* < 0.05) to sexual orientation, religion, cohabitation, and drug consumption before and during sexual intercourse (Table 2).

Among heterosexuals, 82.5% used contraception. This percentage decreased to 71.2% among bisexuals and 53.1% among individuals who identified as gay or lesbian.

In terms of religion, among Catholics, 78.8% said they used contraceptives; this number was 86.0% among agnostics/atheists and 54.5% among those of other religions. A total of 67.1% of those cohabiting used contraceptives compared with 81.4% of non-cohabiting participants.

A total of 78.2% of the participants using contraceptives consumed pornography 1 to 3 times/week, and 1.9% used drugs during sexual intercourse.

#### 3.1.3. History of STIs

A total of 95.2% of participants said they did not have or had never had STIs, 3.6% said they had had STIs, and 1.2% preferred not to say (Table 1). The most common diseases among those who had had STIs were HPV and chlamydia.

There was a significant relationship (*p* < 0.05) between the history of STIs and the type of relationship, the use of dating apps, and the frequency of drug consumption and drug consumption during sexual intercourse (Table 2). Of those who declared that they had or had had STIs, 75% were in an exclusive or conventional relationship, 50% used dating apps, 72.2% used drugs 1–3 times per week, and 11.5% used drugs before or during sexual intercourse.

#### 3.1.4. Use of Dating Applications

A total of 21.1% of respondents indicated that they used dating apps (Table 3). The use of dating apps was significantly related (*p* < 0.05) to gender, sexual orientation, religion, having a partner or not, the type of relationship, the history of STIs, pornography, and drug use (Table 2). In terms of gender, 18.8% of women and 26.8% of men indicated that they used dating apps. In relation to sexual orientation, 59.4% of individuals who identified as gay or lesbian, 44.1% of bisexuals, and 15.6% of heterosexuals indicated that they used these applications. A total of 14.9% of Catholics and 34.5% of agnostics stated that they used dating apps. In total, 30.0% of those without a partner and 14.5% of those with a partner used dating apps. In terms of relationship type, 66.7% of those in an open relationship used dating apps, in contrast to 12.4% of those in an exclusive or conventional relationship. Dating apps were used by 50.0% of those who had or had had an STI, compared to 19.2% of those who had never had an STI. When considering pornography and drugs, 35.7% of pornography users and 24.8% of drug users also used dating apps; these percentages dropped to 11.4% and 16.0% among those who did not use pornography or drugs, respectively.

#### 3.1.5. Pornography Consumption

In total, 58.8% of the sample indicated that they did not consume pornography, 34.9% indicated that they did, and 6.3% preferred not to say (Table 3). Among those who consumed pornography, 76.5% indicated that they consumed it 1–3 times a week, 15.1% consumed it 4–5 times a week, and 8.4% consumed it almost daily.

Significant relationships (*p* < 0.05) were observed between pornography use and gender, sexual orientation, religion, the type of relationship, the use of dating apps, and drug use (Table 2).

In total, 66.0% of men and 22.1% of women indicated that they consumed pornography. In relation to sexual orientation, 53.1% of individuals who identified as gay or lesbian, 47.1% of bisexuals, and 32.3% of heterosexuals said they consumed pornography. If we consider pornography consumption by religion, the percentages were 48.3%, 29.5%, and 36.4% among agnostics/atheists, Catholics, and those practicing other religions, respectively. In addition, 66.7% of those in an open relationship consumed pornography, compared to 31.2% of those in an exclusive or conventional relationship. Among dating app users, 59.1% also consumed porn, while this percentage was 28.5% among those who did not use dating apps.

#### 3.1.6. Drug Consumption

In total, 59.6% of the sample used drugs compared to 40.4% who did not (Table 3). Among those who used drugs, 89.1% used drugs 1–3 times a week, 9.0% used drugs 4–5 times a week, and 1.9% used drugs daily. Drug use was significantly related (*p* < 0.05) to gender, religion, having a partner or not, the type of relationship, cohabitation, the use of dating apps, and porn use (Table 2).

In total, 55.9% of women used drugs compared to 68.4% of men. In relation to religion, 57.4% of Catholics and 64.9% of agnostics/atheists indicated that they used drugs. Among those who used drugs, 51.4% had a partner, and 93.6% indicated that they were in an exclusive or conventional relationship, although 92.8% did not live with their partner, 24.8% used dating apps, and 44.2% used porn.

## 4. Discussion

The main objective of this study was to explore how sociodemographic variables, sexual behaviors, dating app use, drugs, and pornography interact in a population of university health science students. The findings reveal significant relationships between these variables that should be discussed in the context of the existing literature and their implications for young people’s sexual and general health.

### 4.1. Socio-Demographic Variables

The predominance of women in the sample is consistent with the higher representation of women in health science programs. The observed differences in sexual attitudes according to gender, sexual orientation, and marital status underscore the need to take these variables into account when designing sexual health interventions, as has been noted in previous studies [27,28]. In particular, the relationship between gender and the use of dating apps, as well as the consumption of pornographic material, points to differences in how men and women seek and make sexual connections, as other studies have also indicated [29,30,31,32].

### 4.2. Sexual Orientation

Results show that differences in sexual orientation are significantly related to several variables, such as the use of dating apps and the use of pornography and drugs before or during sex. In line with previous research, individuals who identify as gay or lesbian and bisexual people report a higher frequency of dating app use and pornography consumption compared to heterosexuals [29,30,31,32]. This higher frequency may be related to their search for affective and sexual connections on digital platforms, which provide them with a space to explore their sexuality and establish affective and sexual connections in a more accessible, anonymous, discreet, and open way. In many cases, these groups may not easily find these types of connections in their immediate social environment, which increases the importance of digital applications in their social and sexual lives.

The results also indicated differences in contraceptive use by sexual orientation. One possible explanation could be due to the fact that people of different orientations have different needs in terms of pregnancy prevention. For example, heterosexual people tend to have a stronger focus on pregnancy prevention, leading them to use methods such as condoms or birth control pills. By contrast, individuals who identify as gay or lesbian, not being confronted with the risk of pregnancy in their relationships, may prioritize other aspects of sexuality, which may influence their decisions about contraceptive use. In reference to the relationship between drug use and sexual orientation, the results show that individuals who identify as gay or lesbian and bisexual people report higher drug use compared to heterosexuals. This finding aligns with recent scientific evidence suggesting that drug use in sexual contexts (chemsex) is relatively common among certain subgroups within the LGTBI community, particularly among men who have sex with men [33].

Our findings have also indicated that pornography use is more prevalent among individuals who identify as gay or lesbian and bisexual people compared to heterosexuals. This result seems to be in line with different research and even seems to be related to the increase in risky sexual behavior in this group [34].

### 4.3. Use of Contraceptive Methods

In our study, contraceptive use showed a significant relationship to religion and cohabitation and drug use before and during sex. Although some previous studies might suggest that religious people are less likely to use contraception due to doctrinal restrictions, the results of this study suggest otherwise, indicating a pragmatic integration between religious beliefs and reproductive health practices [33]. This finding may reflect greater awareness of family planning and STI prevention among people of faith, highlighting the importance of further exploring these dynamics in future studies.

In addition, our results show that cohabiting partners tend to use contraception less frequently than non-cohabiting partners. This finding could be interpreted by the fact that cohabiting couples may often feel more stable and confident in their relationship [35], which may lead to shared decisions about contraceptive methods, such as the use of more long-term methods (IUDs, sterilization, etc.), or even to the idea that family planning is more controlled and less dependent on contraceptive use during every sexual encounter [36].

Another relationship worth noting involves the use of drugs before or during sex and contraceptive use. According to other research, people who use drugs before or during sex may have a decreased ability to make informed decisions about contraceptive use. Substance use may reduce risk perception, which could lead to reduced use of condoms or other contraceptive methods during sex [37,38].

### 4.4. History of STIs

There was a significant association between the history of STIs and dating app use. These findings are consistent with previous research suggesting that dating app use may increase STI risk due to the increased number of sexual partners and possibly lower frequency of protection [34]. However, it is critical to remember that correlation does not imply causation, and we cannot conclude that dating app use causes an increase in STIs without additional research addressing this issue. Also, the number of students that declared a history of STIs was low; therefore, these results cannot be generalized.

### 4.5. Dating App Use

The use of dating apps is related to a variety of sociodemographic factors, including gender, sexual orientation, religion and relationship type, history of STIs, and porn and drug use. In line with previous studies, men and gay and bisexual people are more likely to use these apps, which may reflect differences in patterns of sexual and affectionate partnership search [29,30,31,32]. In addition, people in open relationships showed a higher frequency of dating app use compared to those in conventional relationships, which may be linked to the nature of non-monogamous relationships and the search for new sexual experiences [39]. The use of dating apps among agnostics and atheists compared to religious people, such as Catholics, may be attributed to different attitudes towards relationships and sexual norms. This divergence could be influenced, as some research indicates, by the cultural and religious contexts that shape users’ behaviors and expectations [40]. Specifically, some authors indicate that higher levels of religiosity are associated with negative attitudes toward sexually explicit material and traditional views about relationships, which may deter believers from using dating apps [41].

The relationship between dating app use and STI prevalence is somewhat more complex. However, there is evidence that shows that people who use dating apps may engage in riskier sexual behaviors, thereby increasing their exposure to STIs. This phenomenon is particularly pronounced among men who have sex with men [42,43,44].

Moreover, as previous research suggests, motivations for using dating apps may play a crucial role in the association between these platforms and both sexual and substance-use related behaviors [45]. Different reasons, such as seeking romantic relationships, exploring one’s sexuality, or the desire for novelty, can influence how individuals engage with these platforms and shape their behavioral patterns [46]. Examining these motivations in greater depth would provide a better understanding of the dynamics between dating app use and its implications for sexual health and overall well-being.

These motivations may also intersect with other behavioral patterns. For instance, previous studies have indicated that individuals who consume pornography, drugs, and alcohol tend to use dating apps more frequently compared to those who do not [45,46]. This pattern may be linked to a more active interest in sexual exploration or the pursuit of sexual satisfaction through various means, both digital and in-person.

Consistent with this, the literature also reports that differences in motivations for using dating apps may play a key role in shaping users’ behavioral patterns [46]. Some studies have suggested that different reasons for engaging with these platforms can modulate the relationship between dating app use and various behaviors, including those related to sexual activity and substance use. Expanding research in this direction would contribute to a better understanding of the factors influencing these associations and support the development of more effective public health strategies.

### 4.6. Consumption of Pornographic Material 

Pornography use is also significantly related to gender, sexual orientation, and relationship type. Men reported a higher frequency of pornography consumption than women, which is consistent with previous research showing gender differences in the access to and use of sexually explicit material [47,48]. In addition, people in open relationships showed a greater tendency to consume pornography, which may be related to more liberal attitudes towards sexuality in this group [49,50].

### 4.7. Drug Use

Our findings on drug use are worrying, as they showed that a significant percentage of the respondents used drugs. Drug use was significantly related to gender, religion, and the type of relationship. Men reported higher rates of drug use than women, which is consistent with previous studies [51]. In addition, the fact that people who live together with their partner show a lower tendency to use drugs may indicate a more stable lifestyle, although this relationship is not entirely clear. Nevertheless, a significant relationship was found between drug use before or during sex and STIs, which underlines the importance of integrating substance use prevention into sexual education programs [52,53].

Although this study found significant relationships between several variables, such as drug use and sexual orientation or the use of dating apps and STIs, in a fairly large population of health sciences students, it is important to keep in mind some important limitations of this study, listed as follows: The sample was limited to students from a certain area and socio-cultural context, which might limit the generalizability of the results to the general population.The data were based on self-report surveys, which may lead to response biases, especially on sensitive topics such as sexuality, drug use, or pornography use. In addition, due to the cross-sectional and voluntary nature of this study, respondents might not fully represent the entire university population, which could introduce a social desirability bias.The cross-sectional descriptive design of this study prevents the establishment of cause-and-effect relationships between variables. It would be interesting in future research to delve more deeply into the significant correlations indicated throughout this study, with the aim of establishing a cause–effect relationship between some of them.Although aspects such as sexual orientation, religion, and sexual attitudes were mentioned, this study could have benefited from a more in-depth study of other psychological factors, such as empathy, personality, and other variables considered important in this type of study.Additionally, by using categorical variables, some nuances in the data might have been lost, potentially limiting the depth of the findings. Continuous measures could provide a more precise understanding of the relationships between variables, capturing variations that may be critical for more refined statistical analyses.Although this study identified significant associations between dating app use and risky behaviors, these findings do not imply a direct causal relationship. Other mediating factors, such as personality traits, peer influence, or underlying psychological variables, may contribute to these behaviors. It would be advisable for future research to explore these mediating factors through longitudinal designs or experimental studies to better understand these relationships.

## 5. Conclusions

Among the main findings of this research, it can be highlighted that the data indicate that, within the sample studied, participants who identified as gay, lesbian, or bisexual reported higher use of dating apps, drug use, and pornography use compared to those who identified as heterosexual. These findings could be related to specific social dynamics or access to digital socialization spaces, which could be explored in future studies. In addition, these groups showed lower adherence to contraceptive and protective methods. Likewise, the high prevalence of drug use among students highlights the need to implement prevention and awareness programs, with special attention directed at the most at-risk groups. These findings emphasize the importance of integrating this knowledge into nursing clinical practice, particularly in the design of educational interventions that promote sexual health, responsible behaviors, and inclusion, thus strengthening the role of nursing in STI prevention and risk behavior management. In terms of interventions to reduce exposure to STIs in the most at-risk populations, it would be essential to implement comprehensive sexual education programs, ensure access to preventive resources such as condoms and pre-exposure prophylaxis (PrEP), and develop awareness campaigns aimed at reducing risky sexual behaviors. In addition, future studies should explore the effectiveness of these strategies through more advanced methodological approaches, which will strengthen public health policies and improve the prevention and management of STIs.

## Guidelines and Standards Statement

This manuscript was drafted against the STROBE guidelines.

## Figures and Tables

**Table 1 nursrep-15-00083-t001:** Sociodemographic characteristics, use of dating apps, and consumption of drugs and pornography in the study sample.

Variable	*n* (%)
Age	21 (20–22)
Gender	
Female	511 (70.0)
Male	209 (28.6)
Other	10 (1.4)
Sexual orientation	
Individuals who identify as gay or lesbian	32 (4.4)
Heterosexual	616 (84.4)
Bisexual	68 (9.3)
Not specified	14 (1.9)
Religion	
Agnostic	203 (27.8)
Catholic	505 (69.2)
Muslim	11 (1.5)
Protestant	11 (1.5)
Having a partner	
Yes	414 (56.9)
No	313 (43.1)
Type of relationship	
Open relationship	18 (4.2)
Exclusive or traditional	410 (95.8)
Cohabitation as a couple	
Yes	71 (9.7)
No	658 (90.3)
Use of contraception	
Yes	574(80.1)
Condoms	376(65.6) *
Contraceptive pills	157(27.4) *
Coitus interruptus	21(3.7) *
Vaginal ring	12(2.1) *
IUD	4(0.7) *
Emergency contraception	2(0.3) *
Female condoms	1(0.2) *
No	143 (19.9)
Sexually transmitted infections	
Yes	26 (3.6)
HPV	4 (15.4) *
Chlamydia	3 (11.5) *
Gonorrhea	2 (7.7) *
Syphilis	1 (3.8) *
Genital Herpes	1 (3.8) *
Not specified	15 (57.7) *
No	688 (95.2)
I prefer not to say	9 (1.2)

Quantitative variable (age) is expressed as median (IQR). Qualitative variables are expressed as frequency (n) and percentage (%); IUD = intrauterine device; HPV= human papillomavirus; * = percent of those answering “Yes”.

**Table 2 nursrep-15-00083-t002:** Results of the Chi-squared test to measure the relation between categorical variables.

	GENDER	SEX O.	RELIG	PARTNER	REL	COHAB.	CONT	STI	APP	PORN	DRUG	DRUGSEX
GENDER	1											
SEX O.	23.56 **	1										
RELIGION	9.51 **	22.11 **	1									
PARTNER	4.04 *	0.81	11.05 **	1								
RELATION	2.53	7.98 *	0.13	19.29 **	1							
COHAB	0.01	2.02	0.04	44.60 **	1.55	1						
CONT.	0.32	20.80 **	5.63 **	0.17	1.94	7.22 **	1					
STI	2.03	1.30	0.67	2.10	16.32 **	1.01	1.09	1				
APP	5.71 *	58.36 **	27.06 **	25.86 **	40.39 **	0.37	0.65	14.70 **	1			
PORN	140.23 **	10.60 **	22.73 **	2.47	9.96 **	1.78	3.46	0.14	59.14 **	1		
DRUG	9.59 **	2.93	3.30 *	10.37 **	5.83 *	8.23 **	0.29	1.09	7.95 **	42.43 **	1	
DRUGSEX	0.90	16.43 **	0.64	0.67	119.77 **	0.17	5.60 *	10.65 *	2.99	8.62 *	10.01 **	1

COHAB = cohabitation as a couple; CONT. = contraception; SEX. O. = sexual orientation; APP = use of dating apps; DRUGSEX = drug consumption before and during sexual intercourse; * indicates a significant correlation, *p* < 0.05. ** indicates a significant correlation, *p* < 0.01.

**Table 3 nursrep-15-00083-t003:** Use of dating apps and consumption of drugs and pornography in the study sample.

Variable	n = 730
Use of dating apps	
Yes	154 (21.1)
No	576 (78.9)
Pornography consumption	
Yes	255 (34.9)
No	429 (58.8)
I prefer not to say	46 (6.3)
Drug consumption	
Yes	432 (59.6)
No	293 (40.4)
Drug consumption before or during sexual intercourse	
Yes	19 (2.6)
No	711 (97.4)

Quantitative variable (age) is expressed as median (IQR). Qualitative variables are expressed as frequency (%); HPV = human papillomavirus.

## Data Availability

The data presented in this study are available on request from the corresponding author. The data are not publicly available due to privacy restrictions.

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
