# Peer review of "Risk Determinants of Sexual Behaviors: Dating Apps, History of Sexually Transmitted Infections, Substance Use, and Pornography Consumption in Health Science Students"

_nursrep, 2025, doi:10.3390/nursrep15030083_

Round 1
Reviewer 1 Report
Comments and Suggestions for Authors
Thank you for the opportunity to review the manuscript “Risk Determinants in Sexual Behaviours: Dating Apps, history of sexually transmitted infections, Substance Use, Pornography Consumption in Health Science Students”.
Overall, this study is valuable and deserves to be published, as it provides relevant information on a large sample of mostly female nursing students who completed a questionnaire on their sexual behavior and related variables. The study was well designed, received ethical approval and presents an updated literature review. Although some references are in Spanish, which may slightly limit its international scope, the conclusions are solid and the limitations are well discussed. From an editorial point of view, it is recommended to use the term “pornography” instead of “porn” to maintain consistency and professionalism.
As an area for improvement, the authors should acknowledge that the use of categorical variables (i.e., yes/no levels) limited the possibility of applying more robust statistical analyses, such as correlation, regression, and mediation, which would have allowed for a deeper exploration of the issues addressed. It is suggested that future studies employ continuous measures to allow for more detailed and enriching analyses.
Finally, based on the findings presented, it is recommended that interventions aimed at reducing exposure to STIs in higher risk populations be implemented.
This could include comprehensive sexuality education programs, increased access to preventive resources such as condoms and PrEP, as well as awareness campaigns focused on reducing risky sexual behaviors. Future studies could explore the effectiveness of these strategies using more advanced methodological approaches to strengthen public health policies.
The study is well organized and designed and presents an up-to-date literature review. The sample is considerably large and is mostly composed of nursing students, which provides significant data on this population.
However, after analysis of the manuscript, I feel that some aspects could be improved to strengthen the validity and impact of the study.
Specific comments on different key points are presented below:
Literature review:
The study addresses a current and relevant topic in the field of sexual health, especially in young and university populations. However, it would be helpful if the authors could more clearly highlight what gap in the literature they are addressing and how their findings complement or extend previous studies.
Methodology
1. The study uses dichotomous categorical variables (yes/no) to analyze risk behaviors, which limits the possibility of applying more robust statistical analyses, such as correlations, regressions or mediation analyses, that would allow a deeper exploration of the relationship between the variables studied. It is recommended that the authors explicitly acknowledge this limitation and suggest the use of continuous measures in future studies.
2. Although the sample size is adequate, more details on the recruitment of participants could be included to ensure the representativeness of the data and avoid selection bias.
3. The use of self-administered online surveys may introduce social desirability or self-selection biases, which could influence the results. It would be advisable for the authors to mention this in the limitations section.
Tables
It is recommended that the authors revise the variable names in the tables in order to homogenise them. In table 2, a variable called Sexual Orientation appears in the rows, while in the columns the same variable is called O.sex.
Written Expressions of Results
In several sections of the manuscript, differences in dating app use, pornography use, and drug use between different sexual orientation groups are highlighted. The results are based on a descriptive analysis of the sample collected; however, the way they are expressed may suggest a generalization of certain behaviors.
Would consider re-phrasing some phrases. Instead of stating that “homosexual and bisexual individuals have significantly higher rates of dating app use and substance use,” it could be rephrased with a more neutral, data-driven approach, for example:
“The data indicate that, within the sample studied, participants who identified as homosexual or bisexual reported higher use of dating apps and substance use compared to those who identified as heterosexual. These findings could be related to specific social dynamics or access to digital socialization spaces, which could be explored in future studies.”
This rephrasing would avoid biased interpretations.
Conclusions
The conclusions of the study are sound and well-grounded in the data. However, it is suggested to emphasize that the relationship between dating app use and risk behaviours does not imply direct causality, but may be mediated by other factors, perhaps including this part in limitations or future interventions could be useful.
Finally, it is advisable to maintain a more academic and professional tone, such as replacing the term “porn” with pornography.
Author Response
Dear Reviewer,
We sincerely appreciate the opportunity to submit our manuscript for review and your interest in our work and its potential publication. In response to your valuable feedback, we have carefully reviewed and addressed each of your comments, implementing the necessary revisions to enhance the quality and clarity of the article. Please find attached a document detailing our responses to all comments.

Reviewer 2 Report
Comments and Suggestions for Authors
Thank you for the opportunity to review this article, dealing with an interesting topic.
Although the timely issue of the paper, I think some relevant issue should be considered and amended before acceptance and publication of the article.
Introduction: the authors dedicate a large section of the introduction to the topic of STIs, focusing on medical aspects. Given the focus of the paper, I suggest to widen the discussion of previous studies regarding the other considered variables.
Methods: it is not clear the recruitment method of the sample; in the method section the authors indicate "convenience sample". In the title they indicate "health science students".
Data analysis and results: the statistics used seem very simple. I suggest the authors to use a regression model to investigate the relation among independent and dependent variables
Table 1: in the table the authors only indicate Yes/No as answers to the "Use of Contraception". In the text it comes out that other answers were asked (condoms, contraceptive pills, other?). In the same table I can't find information about drug consumption and pornography consumption.
With the only aim to ameliorate the quality of the paper I suggest the authors to consider these suggestions.
Author Response

(The authors gave the same response as above.)

Round 2
Reviewer 2 Report
Comments and Suggestions for Authors
The revised version of the paper provides a higher consistency in the presentation of the paper.
In consideration of the added paragraph and the discussion section, I suggest the authors to make some reference in regard of previous research investigating the relationship between dating app use and substance-related behaviors (i.e. https://doi.org/10.3390/socsci10070249 )
The authors, in the discussion section, could also consider the role of motivations for using dating apps, that previous research have indicated to modulated the association between dating app use and both sexual and substance-related behaviors.
Author Response
Dear Reviewer,
Thank you very much for your valuable suggestions. We have carefully considered your recommendations and have expanded the discussion section accordingly. In particular, we have included references to the suggested research, as well as other relevant studies that we consider necessary to enhance the quality of our manuscript and open new avenues for further research on this topic (references 45-47).
Furthermore, we have added a dedicated section on the role of motivations in the use of dating applications, acknowledging their influence on the association between dating app use and both sexual behaviors and substance-related behaviors, as highlighted in previous research (lines 355-373).
We sincerely appreciate your insightful comments, which have significantly contributed to improving our work.
